# Research on the Heterogeneity Threshold Effect of Foreign Direct Investment and Corporate Social Responsibility on Haze Pollution

**DOI:** 10.3390/ijerph20064802

**Published:** 2023-03-09

**Authors:** Zhanjie Wang, Yongfeng Ma, Shasha Wang, Yongjian Wang

**Affiliations:** 1School of Business Administration, Guizhou University of Finance and Economics, Guiyang 550025, China; 2Graduate School, Guizhou University of Finance and Economics, Guiyang 550025, China; 3Business School, Jiangsu Normal University, Xuzhou 221116, China

**Keywords:** foreign direct investment, corporate social responsibility, haze pollution, threshold effect, heterogeneity analysis, fixed-effect model

## Abstract

Carrying out environmental protection and governance in the process of using foreign capital to develop the economy is a realistic problem that China needs to solve urgently. In order to reduce environmental pollution, all enterprises are called upon by the local government to fulfil CSR and improve the quality of FDI use. However, previous studies have rarely explored the threshold effect of FDI and CSR on haze pollution. This paper employs the threshold effect model to explore the above problem based on panel data of 30 provinces in China from 2009 to 2018. The empirical study found the following: (1) FDI has a significantly positive double-threshold effect on haze pollution. Meanwhile, the promotion effect of FDI on haze pollution is the strongest in the two threshold ranges. (2) CSR has a significantly negative single-threshold effect on haze pollution; that is, the increase in CSR intensity inhibits haze pollution. Such a negative effect shows the characteristics of increasing marginal efficiency. (3) In addition, the provinces in different thresholds display obvious geographical distribution characteristics. Through the above analysis, it can be observed that FDI and CSR have distinct impacts on haze pollution. Thus, the country and the government can reduce haze pollution by improving the investment structure, using environmentally friendly technology, guiding enterprises to abide by business ethics and promoting social responsibilities fulfilment.

## 1. Introduction

Since the reform and opening up, foreign direct investment (FDI) has played a vital role in promoting the economic development and optimizing the foreign trade structure of China. However, whilst boosting the economic strength of the country, FDI has also brought significant ecological damage and environmental pollution [1]. Using the Beijing–Tianjin–Hebei region as an example, according to China Statistical Yearbook data, the amount of FDI increased from USD 18.15 billion to USD 48.43 billion, with an increase of 166.8% during the period of 2009–2018. Meanwhile, the GDP in the same period increased from USD 484.43 billion to USD 11285.7 billion, an increase in more than 20 times. It can be seen that the GDP of the Beijing–Tianjin–Hebei region has also achieved continuous growth with the increase in the number of FDI. However, according to the Announcement of China’s Environmental Status in 2018 issued by the Ministry of Environmental Protection, the number of days of light pollution, moderate pollution, heavy pollution and serious pollution in Beijing–Tianjin–Hebei region accounted for 27.1%, 10.50%, 6.0% and 3.20%, respectively, in a year. The announcement showed that the average concentration of PM 2.5 is 77 μg/m^3^, 1.20 times higher than the national secondary standard [2].

Driven by industrialization, western countries have produced massive amounts of air pollution for over a century that have affected the economic development of China over the past 40 years. The frequent occurrence of haze pollution events, including PM2.5, PM10 and other significant sources of pollution, has gradually expanded the scope and degree of pollution. Therefore, many scholars have proposed the ‘pollution paradise’ effect of FDI. To circumvent environmental regulations in their own countries and transfer crude industries, some multinational enterprises take advantage of their capital to export highly polluting and energy-intensive industries to less developed countries that are in urgent need of economic development and have low environmental awareness, thereby becoming essential drivers of environmental degradation [3,4,5]. Liu and Gao highlighted a strong positive correlation between FDI and environmental pollution, arguing that the degree of regional pollution is aggravated along with an increasing FDI agglomeration [6]. Dong et al. confirmed the positive effect of FDI on haze pollution based on quantile regression and Shapley value decomposition [7].

However, some scholars also put forward the ‘pollution halo’ effect of FDI. Less-developed countries that introduced advanced clean production technologies and environmental governance from developed countries through FDI channels could improve their utilization rate of natural resources and the quality of their ecological environment [8,9]. He and Liu concluded that in China, FDI had a positive effect on pollution emissions (especially industrial sulfur dioxide emissions), and the impact of FDI on environmental pollution varied significantly across the eastern, central and western regions of China [10]. Nathaniel et al. pointed out that the hypothesis of FDI effect was not valid in Mediterranean coastal countries, but FDI could effectively promote the improvement of local environmental quality [11]. In response to these contrasting findings, some scholars have explored the green technology spillover effects of induced labor- and capital-based FDI [12]. Some scholars have taken another approach and analysed the problem from the perspective of the ‘coordinated development of two-way FDI’, concluding that the coordinated development of two-way FDI in China could significantly suppress haze pollution [13]. However, a unified conclusion on this topic is yet to be reached. This paper approves the ‘pollution paradise’ effect of FDI because the high incidence of haze pollution not only affects the health and well-being of the people, but also poses a huge threat to the ecological civilization construction and low-carbon green growth in China [14]. Therefore, how to encourage countries to commit themselves to increasing their levels of environmental protection whilst simultaneously boosting trade and investment via FDI has become a hot topic amongst scholars.

Very few scholars have explored the impact of CSR on haze pollution. As micro-entities of the regional economy, enterprises serve as creators of economic value and producers of environmental pollution. Their functional performance in their economic, social and environmental responsibilities contributes to improving regional environmental quality and overall social welfare [15,16]. In this sense, encouraging more companies to commit themselves to social and environmental issues can help societies increase their trust in business communities, enhance the social capital of companies and thereby lead to a synergistic win-win situation [17]. In terms of CSR transmissibility, if core firms in a region are active in haze reduction and management, then their activities will significantly affect the ecosystem of the industrial chain [18]. The core enterprises set an example of continuously making significant contributions to the overall environment, especially haze pollution, by promoting social responsibility awareness on the upper and lower levels of the industrial chain. Therefore, if more firms voluntarily commit to limiting their pollution emissions, even beyond the provisions of international protocols and treaties, then such environmentally responsible behavior can become a benchmark for competitors to follow, thus forming a virtuous catch-up cycle [19]. Some scholars have also pointed out that the CSR activities vigorously carried out by enterprises can stimulate active cognitive responsibility feedback from the society. Enterprises can attract those who care about social and environmental issues to act together by engaging in green and low-carbon production and designing environment friendly and energy efficient products to inform consumers that the production process of their final products minimizes harm to the environment [20]. Such behavior can also help consumers be generally aware that companies, even sometimes seen as purely economic actors, have made environmentally responsible commitments in this area [21].

Furthermore, in the context of the influx of FDI into China’s economic construction, FDI may indirectly influence the level of haze pollution in China through corporate social responsibility (CSR). Foreign companies with high CSR compliance standards may gradually lower their compliance standards whilst adapting to the already tricky and ineffective CSR compliance situation in China [22], thereby forming a vicious circle of ‘competition to the bottom’ with domestic companies [23]. Low-quality CSR activities vastly reduce discriminatory barriers and the transparency of international production activities. These activities also result in the indifference of foreign companies towards the welfare of Chinese consumers, the greening of production processes and their fulfilment of environmental responsibility. The resulting ‘pollution paradise’ effect will also further aggravate the degree of haze pollution [24] and ultimately reduce the overall level of CSR and further aggravate the environmental pollution in China. During the investment process, enterprises need to be jointly driven to fulfil their social responsibilities, strengthen their environmental awareness and promote a balanced development of bilateral economy, ecology and society through moral guidance and environmental regulations.

To cope with the further deterioration of the ecological environment, China actively advocates optimizing the structure and quality of its FDI, enhancing CSR fulfilment and promoting green technology innovation [25]. Nevertheless, the balance in the relationship amongst FDI, CSR and haze pollution remains a practical problem that hinders the low-carbon green development of the country. On the basis of the general equilibrium model of Copeland and Taylor, most domestic and foreign scholars have investigated the internal relationship amongst FDI, CSR and environmental pollution under the constraints of FDI and environmental regulation conditions [15,26]. In terms of research methods, scholars have transitioned from the econometric model with ordinary least square method and simultaneous equations at the core to the endogenous growth model and data envelopment analysis [26,27]. On the basis of the regional differences in practice and development, much achievement has been reported on spatio-temporal evolution analysis and sub-regional testing via micro-scopization [28,29]. In the context of technological innovation leading to low-carbon and green development, individual scholars have introduced environmental technology innovation behavior based on the ‘factor–behavior–performance’ research idea to understand the role of FDI, environmental regulation and technology innovation in environmental performance paths [30]. To address this problem, this study attempts to integrate FDI, CSR and haze pollution into a unified theoretical framework, explore the nonlinear effects of FDI and CSR on haze pollution, further explore the problem based on regional economic level and resource endowment heterogeneity, and provide a scientific and sound theoretical basis for the haze pollution management and ecological environmental protection decisions in each province.

The theoretical significance of this paper is as follows: Firstly, in the issue of the CSR measurement, this paper innovatively proposes an optimized method that considers the social responsibility carrying capacity of enterprises of different scales and regions and the matching degree of local economic development. Through theoretical deduction and empirical results, it is preliminarily confirmed that CSR has an inhibition function on haze pollution in terms of time and depth, and the research findings provide a reference for relevant theory. Secondly, the action path and boundary conditions of FDI and CSR on haze pollution are proposed based on panel threshold model. This can objectively evaluate the environmental effects of FDI and provide a new perspective for identifying the drivers and governance mechanisms of haze pollution. Thirdly, this paper takes FDI and CSR as threshold variables, preliminary fits the findings through panel regression and analyses the spatial–geographical distribution characteristics of provinces based on different interval thresholds, thereby further verifying the completeness of its findings.

The remainder of the paper is presented as follows: Section 2 provides an overview of methodology and data. Section 3 represents research results and analysis. Section 4 reports the discussion, and Section 5 reports the conclusion.

## 2. Methodology and Data

### 2.1. Theoretical Mechanisms and Research Hypotheses

FDI exerts a driving effect on haze pollution mainly through the technology-locking and extrusion effects. The technology-locking effect is manifested in the transfer of high-value industries and the export of clean technologies. The transfer of high-value industries from developed countries brings a large amount of capital to less-developed countries and regions, boosts economic growth and, to some extent, provides financial support for local ecological and environmental pollution management [12,19]. However, the inflow of large amounts of foreign capital can lead to R&D inertia in less-developed countries and regions, thereby curbing the enthusiasm of domestic enterprises in scientific research and innovation and stagnating the number of green capital investments and patent applications, thus leading to the ‘technology-locking’ phenomenon [31,32]. Meanwhile, developed countries will not directly send their most cutting-edge clean technologies to less-developed countries and regions because they need to maintain their monopoly over clean technology. In addition, the technological gap between less-developed countries and regions and the limitations of many factors, such as digestion and absorption capacity [27,33], eventually triggers a ‘technology-locking’ effect in these areas. Given the crowding-out effect, the use of FDI from ‘quantity-to-quality’ momentum conversion is driven by national policy and institutional guidance. To give full play to the economic, technological and environmental effects of FDI, the government formulates various policies and regulations to regulate the market, which reduce its expenditure in other fields [1,32]. Meanwhile, under the thinking of local government officials who are eager for quick success and immediate benefits, and the thinking of only GDP-oriented growth and the current performance appraisal system, local government still takes FDI with solid liquidity and fast returns as their first choice to attract investment [30,31] and excludes those FDI with high environmental protection, clean requirements and a low conversion rate of technological achievements, thereby falling into a vicious circle of pursuing FDI quantity [34], which will eventually aggravate the level of regional haze pollution. Hypothesis 1 is then proposed, that is, that there is a nonlinear double-threshold positive effect between FDI and haze pollution.

CSR takes the spillover and leverage effects as carriers to inhibit haze pollution. Previous studies show that with the improvement of CSR performance, low carbon green development can promote regional ecological environment quality by improving the CSR performance intensity in the region [35]. Although CSR fulfilment increases capital costs in the short term and is not conducive to improving economic performance, CSR fulfilment can promote the customer-oriented strategy of enterprises and improve their business performance in the long run [36]. Therefore, enterprises must be committed to CSR activities through various channels and approaches in a specific country or region and strive to build a ‘community of social responsibility with a shared future’. In the form of point-to-line, line-to-surface and surface-to-field connections, and through social responsibility scale effect at the industry level, enterprises balance and address environmental, social and economic benefits to meet the broader social needs for sustainability, including the protection of natural assets, services and functions of ecosystem on which human society ultimately depends [37], thus effectively promoting low-carbon and green development and reducing pollution. The leverage effect is reflected in how, through the formulation and implementation of social strategies, enterprises establish social cooperation and win-win relationship networks with various stakeholders, pool their knowledge to scientifically design and develop low-carbon green products, improve their production process reengineering, promote the use of clean production technologies, reduce their emission of haze pollutants and provide more health products and services for the public [38]. CSR performance also brings economic, social and environmental advantages [33]. Therefore, one can reasonably assume that those enterprises effectively carrying out their social responsibility can maximize their ecological innovation to improve atmospheric conditions and enhance the contributions of their commercial activities to the society and environment [39]. Hypothesis 2 is then proposed, that is, that CSR has a nonlinear double-threshold negative influence on haze pollution.

### 2.2. Variable Definitions

(1)Haze pollution

Given that haze concentration in China was only monitored in 2012, following the bounds of data selection years in this paper and the views of Yan and Qi and Beatriz et al. [8,15], the satellite observation and chemical migration model from the Center for International Earth Science Information Network (CIESIN) at Columbia University in New York City, USA, which was used to export and convert the data into global PM2.5 average annual concentration monitoring raster data. The monitoring results from this data are roughly the same as those from the domestic environmental protection department, thereby verifying their credibility and applicability.

(2)Corporate social responsibility (CSR)

CSR data were collected from the ‘Social Responsibility Report of Listed Companies’ published between 2009 and 2018 by Rankins CSR Ratings, RKS [39]. This report measured the degree of CSR fulfilment from four aspects, namely overall, content, technical and industry. Considering the differences in the CSR fulfilment capacities of different regions and industries, the CSR scores of each enterprise were revised using the enterprise scale from the CSMAR database, that database is developed by Xishma Data Technology Co., Ltd., Shenzhen city, China. Afterwards, the average social responsibility scores of each province and region across different years were calculated and used as evaluation indices of regional social responsibility.

(3)Foreign direct investment (FDI)

The FDI data used in this study included the inflow of capital and technology, the transfer of industries and the absorption and utilization of actual capital. In this paper, the actual amount of foreign capital utilization in each province and city was used to measure the FDI for each year. Given the variability of statistical units, the data were converted based on the USD:RMB exchange rate for the current year to obtain the value of FDI in each province and city, and then the logarithm was taken as the measurement variable [5,10].

(4)Control variables

① Environmental regulation (ER). A greater number of environmental regulations and a stricter degree of regulation correspond to a higher motivation for enterprises to invest in environmental pollution control. In this paper, the ratio of the amount of investment in industrial pollution control to the industry-added value in a specific year was used as a proxy for ER [1]. ② Economic growth (EG). In general, a faster economic growth corresponds to a greater demand for FDI and a greater value of GDP per capita. The GDP per capita of each province and city was then used to measure the level of economic growth [4]. ③ Energy structure (ES). China is currently at the stage of high-quality economic development, and its energy consumption is undergoing structural adjustment from coal to clean energy consumption. Therefore, the proportion of natural gas consumption to total energy consumption was used in this paper as a proxy for ES [40]. ④ Industrial structure (IS). Given that the development of primary and secondary industries has a significant impact on the generation of haze pollution, the ratio of the added value of primary and secondary industries to the GDP of each province and city was used in this paper to measure IS [1]. ⑤ Regional innovation ability (IA). Regional innovation focuses on environmental protection, sustainable development of the society and the degree of regional innovation cannot be precisely measured by the number of patent applications or authorizations alone. Therefore, the strength of provincial and municipal innovation ability was measured in this paper using the comprehensive index of IA following the recommendations from the China Regional Innovation Ability Evaluation Report [41].

### 2.3. The Model

Based on the interaction mechanism amongst the variables, the following panel regression model was constructed to verify the effect of FDI and CSR on haze pollution:(1)$$PM_{\mathrm{it}}=\beta_{0}+\beta_{1}FDI_{it}+\beta_{2}FDI_{\mathrm{it}}^{2}+\beta_{3}CSR_{it}+\beta_{4}EG_{it}+\beta_{5}ES_{it}+\beta_{6}IS_{it}+\beta_{7}ER_{\mathrm{it}}+\beta_{8}IA_{it}+\varepsilon_{it}$$
(2)$$PM_{\mathrm{it}}=\beta_{0}+\beta_{1}CSR_{it}+\beta_{2}CSR_{{}_{it}}^{2}+\beta_{3}FDI_{it}+\beta_{4}EG_{it}+\beta_{5}ES_{it}+\beta_{6}IS_{it}+\beta_{7}ER_{\mathrm{it}}+\beta_{8}IA_{it}+\varepsilon_{it}$$

The threshold effect model proposed by Hansen was then used to further examine the difference in the fluence degree of FDI and CSR on haze pollution across different threshold domains due to the existence of threshold values. FDI and CSR were taken as threshold variables in turn to establish the following single-threshold effect models for accurately capturing the critical values of explanatory variables and for understanding the nonlinear relationships when the structure changes as follows:(3)$$PM_{\mathrm{it}}=\beta_{i}+\beta_{1}CSR_{it}+\beta_{2}EG_{it}+\beta_{3}ES_{it}+\beta_{4}IS_{it}+\beta_{5}ER_{it}+\beta_{6}IA_{it}+\beta_{7}FDI_{\mathrm{it}}\cdot I(FDI\leq \gamma )+\beta_{8}FDI_{it}\cdot I(FDI>\gamma )+\varepsilon_{it}$$
(4)$$PM_{\mathrm{it}}=\beta_{i}+\beta_{1}FDI_{it}+\beta_{2}EG_{it}+\beta_{3}ES_{it}+\beta_{4}IS_{it}+\beta_{5}ER_{it}+\beta_{6}IA_{it}+\beta_{7}CSR_{\mathrm{it}}\cdot I(CSR\leq \delta )+\beta_{8}CSR_{it}\cdot I(CSR>\delta )+\varepsilon_{it}$$

Given that action may have a double-threshold or even a multi-threshold effect due to the existence of multi-stage characteristics, the above single-threshold model was further extended as follows:(5)$$\begin{array}{l} PM_{\mathrm{it}}=\beta_{i}+\beta_{1}CSR_{it}+\beta_{2}EG_{it}+\beta_{3}ES_{it}+\beta_{4}IS_{it}+\beta_{5}ER_{it}+\beta_{6}IA_{it}+\beta_{7}FDI_{\mathrm{it}}\cdot I(FDI\leq \gamma_{1})+\beta_{8}FDI_{it}\cdot I(\gamma_{1}<FDI\leq \gamma_{2})\\ +\beta_{9}FDI_{\mathrm{it}}\cdot I(FDI_{\mathrm{it}}>\gamma_{2})+\varepsilon_{it} \end{array}$$
(6)$$\begin{array}{l} PM_{\mathrm{it}}=\beta_{i}+\beta_{1}FDI_{it}+\beta_{2}EG_{it}+\beta_{3}ES_{it}+\beta_{4}IS_{it}+\beta_{5}ER_{it}+\beta_{6}IA_{it}+\beta_{7}CSR_{\mathrm{it}}\cdot I(CSR\leq \delta_{1})+\beta_{8}CSR_{it}\cdot I(\delta_{1}<CSR\leq \delta_{2})\\ +\beta_{9}CSR_{it}\cdot I(CSR_{it}>\delta_{2})+\varepsilon_{it} \end{array}$$
where i represents different provinces; t represents different years; $\beta_{0}$ reflects the individual effect of provincial differences; I(•) is the indicator function; $\varepsilon_{\mathrm{it}}$ is the random disturbance term; r and δ are the threshold values of FDI and CSR, respectively; $\beta_{i}$ is the regression coefficient of each variable; PM, FDI and CSR are the explanatory variables (haze pollution) and threshold variables (FDI and CSR) of the study design; and EG, ES, IS, ER and IA are a set of control variables. After estimating the regression coefficients and the corresponding thresholds for each variable, the significance and authenticity of the thresholds were tested. The regression coefficients and corresponding threshold values were obtained based on the minimum sum of squared residuals of the variables under the given threshold number, and the existence of the threshold effect was tested according to the p-value. The consistency of the obtained threshold values with the actual values was evaluated using the likelihood ratio (LR) statistic.

### 2.4. Data Sources

The panel data of 30 provinces and cities in China (excluding Hong Kong, Macao, Taiwan and Tibet) from 2009 to 2018 were taken as the sample. EG, FDI and IS data were collected from China Statistical Yearbook and China Urban Statistical Yearbook; ES and ER data were obtained from the China Energy Statistical Yearbook, China Regional Economic Statistical Yearbook and China Environmental Statistical Yearbook; and IA data were derived from the China Regional Innovation Ability Evaluation Report compiled by the China Science and Technology Development Strategy Research Group and the China Innovation and Entrepreneurship Management Research Center of the University of Chinese Academy of Sciences University. Moreover, CSR data were collected from the ratings released by Rankins CSR Ratings, RKS, which were sorted by the authors into provincial and municipal data, and the PM data were collected from the raster data published by the Center for Socioeconomic Data and Applications of Columbia University based on the annual mean global PM2.5 concentrations monitored by satellites.

## 3. Results

### 3.1. Collinearity Analysis and Model Selection

Firstly, correlation analysis and variance inflation factor (VIF) were used to verify the multicollinearity amongst the variables. As shown in Table 1, none of the correlation coefficients amongst the variables exceed 0.7, thereby indicating that the correlation coefficients are within a reasonable value range. However, the maximum VIF of 2.41 is much smaller than the reference value of 10, thereby confirming the absence of any severe collinearity amongst the variables. Secondly, F-test and Hausman test were performed to determine the most appropriate estimation method for the model. The panel data estimation methods of OLS regression and random effects were rejected by comparative analysis, thereby confirming that the fixed-effects model was suitable for this study.

### 3.2. Panel Regression Analysis

The effects of FDI and CSR on haze pollution were initially estimated using Stata 15.1 to fit the panel data with the fixed effects. To further reveal the differential effects produced by the strength of the explanatory variables on the explained variables, the effects of the first and second powers of FDI and CSR on haze pollution were examined in the model after introducing control variables. Table 2 presents the test results. When the explanatory variable is FDI, its first power regression coefficient is negative and significantly correlated at the 1% level (β = −25.560, *p* < 0.1), whereas its second power coefficient is positive and passes the 5% significance level test (β = 0.645, *p* < 0.05). Therefore, FDI and haze pollution have a U-shaped relationship. With the continuous increase in the total amount of FDI, the haze pollution degree initially decreases and then increases. This trend shows prominent stage characteristics and is constrained by the effect strength of FDI. When CSR is the explanatory variable, the first power coefficient is significantly positive (β = 31.601, *p* > 0.1), whereas the second power coefficient is significantly negative (β = −15.202, *p* < 0.1), thereby suggesting that CSR and haze pollution have an inverted U-shaped structure. As the degree of CSR fulfilment increases, the degree of haze pollution decreases.

### 3.3. Threshold Test

The above panel regression analysis reveals that both FDI and CSR have significant nonlinear effects on haze pollution, which reflects that the different influences of FDI and CSR obviously restrict the degree of haze pollution. Therefore, Bootstrap repeated sampling was performed 300 times to obtain the F statistic and *p* value as well as the corresponding critical value distribution. Table 3 presents the results. Firstly, the results reveal that both the single and double thresholds of FDI are significant at the 1% level. However, the presence of the triple threshold is not significant, thereby implying that this threshold is invalid. In other words, FDI has a significant double-threshold effect on haze pollution. Similarly, only the single threshold of CSR is significant at the 1% level, and the double and triple thresholds are not significant. In other words, CSR only has a single-threshold effect on haze pollution. Secondly, the threshold value was tested to confirm if it is equivalent to the actual value. Table 3 shows that the two thresholds for FDI are 24.877 and 25.558, and the single threshold for CSR is 1.879. To further verify whether the corresponding estimated threshold value was equal to the actual value, the LR statistic was used to draw the likelihood ratio function graph of each threshold value of FDI (Figure 1) and CSR (Figure 2) under the 95% confidence interval. According to the relationship between the actual LR statistic (lowest point) and the critical value (7.35) at the significance level of 5%, the lowest point of LR statistic is significantly lower than 7.35, thereby confirming the consistency between the threshold value of FDI and CSR and the actual value.

### 3.4. Analysis of Threshold Effect

Table 4 presents the results of the threshold regression of FDI and CSR on haze pollution. The threshold effect of FDI was initially evaluated. In general, the positive effect of FDI on haze pollution demonstrates the ‘pollution paradise’ effect because the current use of FDI in China emphasizes quantity over quality. In the context of intensified local competition, local governments blindly expand their use of FDI to promote economic growth. However, those specific industries into which FDI flows are loosely regulated, thereby leading to many FDI flows into industries with high pollution and energy consumption. This behavior also results in the ‘market theft’ effect of FDI, which further deepens the degree of local haze pollution. When the FDI intensity is lower than 24.877, the impact coefficient is 1.6889, which is significant at the 5% level. Under the early extensive economic development mode, FDI influx plays a special role in promoting environmental pollution. In other words, FDI has a significant positive impact on haze pollution within the first threshold. When the FDI intensity ranges between 24.877 and 25.558, the influence is 2.273 at the significance level of 1%, thereby suggesting that during a process of economic growth that relies on FDI for a long time, the transfer of heavily polluting industries and obsolete technology from developed countries further deepens the environmental deterioration of developing countries. In other words, within the two threshold intervals of FDI, the promoting effect of FDI on haze pollution is significantly enhanced. When the FDI intensity is more significant than 25.558, the impact coefficient decreases to 1.637, which also passes the test at the 5% significance level. Compared with the influence strength of the above two threshold intervals, the positive influence of FDI on haze pollution is the weakest after crossing the second threshold value, thereby suggesting that with the increasing demand for high-quality economic development, both the government and enterprises will adjust their use of FDI, upgrade their industrial structure and use part of their funds for environmental governance. Given that the optimization and adjustment of FDI use are part of a long-term process, the transformation from ‘quantity to quality’ has not yet achieved the effect of restraining environmental pollution. After FDI crosses the second threshold value, the influence of FDI on haze pollution becomes positive, but its degree is the weakest. Therefore, hypothesis 1 is verified.

The threshold effect of CSR was then analysed. Overall, CSR shows a negative nonlinear effect on haze pollution, and this negative effect is characterized by increasing marginal efficiency. When pursuing economic value, enterprises actively practice social responsibility by paying attention to social harmony and ecological environmental protection, all-around development of environment friendly products and multi-channel innovation of low-carbon green production technology. As a starting point for the sustainable development of enterprises and a support point for the healthy development of the social economy, CSR plays a crucial role in scientific and technological innovation, industrial adjustment and environmental protection, providing a kinetic conversion for haze pollution. When CSR intensity is lower than 1.879, its impact coefficient on haze pollution is −9.652, which is significant at the 10% level. In other words, at the early stage of economic development, the collaborative economic, social and environmental nature of CSR effectively suppress the level of environmental pollution, thereby verifying that CSR is an excellent path to reduce environmental pollution. Meanwhile, when the CSR intensity is higher than 1.879, its impact on haze pollution is significantly more substantial with an impact coefficient of −14.040, which is significant at the 1% level. This finding can be mainly ascribed to the fact that with the increasing seriousness of environmental pollution, the government gradually increases the strength of its environmental regulations, thereby highlighting the necessity and comprehensiveness of CSR performance. Many enterprises incorporate CSR into their development strategies in response to the calls of the government and the public, regard the International Social Responsibility Guide (ISO26000) as a benchmark and fulfil their CSR by reducing their degree of environmental pollution. Therefore, on both sides of the single-threshold value of CSR, the influence of CSR on haze pollution is negative and gradually enhanced. Hypothesis 2 is then verified.

### 3.5. Further Analysis

There are also changes in the number of provinces across different threshold intervals. According to the different threshold values of FDI and CSR, the 30 provincial samples were divided into 5 intervals to analyse their threshold variability. According to the statistical results in Table 5, FDI and CSR evolve in the direction of adjustment and optimization. From 2009 to 2018, the number of provinces with two variables that are less than the first threshold value showed a downward trend, whilst the number of provinces with two variables that are greater than the first threshold value showed an upward trend. A total of eighteen provinces did not pass the first threshold for FDI in 2018, and these provinces were mainly located in the less-developed regions of central and western China. In the same year, only three provinces passed the second threshold, namely Tianjin, Guangdong and Jiangsu Province. In terms of CSR, due to the influence of China’s economic development model and the degree of CSR fulfilment, none of the provinces passed the first threshold from 2009 to 2011. With the transformation of economic development and the popularization of the social responsibility concept, the number of provinces crossing the first threshold gradually increased since 2012, but their number remains relatively small. These results suggest that the development of China’s economy and its process of environmental governance are accompanied by the gradual optimization of the FDI structure and the development and implementation of CSR.

An analysis of the geographical characteristics of the number of threshold provinces is also included. The number of provinces where FDI and CSR crossed the threshold shows prominent eastern and western geographical characteristics. From the perspective of FDI, those provinces that exceeded the second threshold in 2009 included Shanghai, Shandong, Guangdong, Jiangsu, Zhejiang and Liaoning. Most of these provinces are located in the eastern coastal region, which is favorable to FDI. Some central provinces such as Sichuan, Anhui, Jiangxi, Henan, Hubei and Hunan are located between the two thresholds due to the constraints in resource endowment and industrial transfer. By contrast, fifteen provinces, including Yunnan, Inner Mongolia, Ningxia, Shanxi, Guangxi, Xinjiang and Guizhou, were less than the first threshold due to geographical location and ecological environment constraints. From the CSR perspective, China still has a long way to go to fulfil its social responsibility. Therefore, none of its provinces passed the first threshold before 2011. According to the measurement standard, it was only after 2012 that the CSR value of Yunnan, Shanghai, Sichuan and Guangdong gradually crossed the first threshold, which, to some extent, indicates that the current social responsibility governance work in China remains challenging and requires further planning and promotion.

## 4. Discussions

Healthy investment in China is the guarantee of stable economic growth. Different from the developed countries who adopt industrial development model, developing countries have a larger effective utilization gap of capital, more environmental barriers to hurdle and a more imperfect social responsibility environment. As a result, protecting the environment in China and hindrance factors affecting haze pollution are worth studying. This article highlights the economic–social environmental impact, embodied in FDI and CSR.

For FDI, whether it is linear regression or threshold effect regression, the promotion effect of FDI on haze pollution exists. These results not only support most of these previous studies suggesting the linear influence of FDI on haze pollution from the perspective of the pollution paradise and pollution halo effects [4,5,6], but also expand its nonlinear relationship with stage characteristics. This may be because China attaches importance to the quantity rather than the quality of FDI due to its eagerness to develop the economy, as well as that the relevant environmental protection mechanism and regulatory mechanism are not sound enough, resulting in the consequences of increasing economic aggregate and environmental pollution.

For CSR, we consider the social-responsibility-bearing capacity of enterprises with different scales and from regions and the matching degree of local economic development. Then, our study shows that CSR has a significant inhibitory effect on haze pollution [13,15], whether linear or threshold effect. This is consistent with theoretical deduction, indicating that CSR effect plays a larger part in the threshold affection. It is possible that China’s policies towards CSR have gradually increased, and thus, enterprises vigorously implement social responsibility and nurture the concept of low-carbon green development. When foreign capital can adapt to China’s institutional environment, a harmonious road between economic development and a beautiful environment may take shape.

When taking FDI and CSR as threshold variables, examining the spatial–geographical distribution characteristics of provinces based on different interval thresholds, the study has found that the number of provinces with FDI and CSR greater than the first threshold has gradually increased, indicating that the Chinese government has gradually attached importance to optimizing and adjusting the structure of FDI use and promoting CSR implementation over time.

## 5. Conclusions

By analysing the influence mechanism of FDI and CSR on haze pollution, this paper reveals a nonlinear relationship between FDI and haze pollution based on the panel data of 30 provinces and cities across China from 2009 to 2018 and by using the fixed-effects model and threshold regression analysis. This study also comprehensively examines the change in the number of provinces based on the threshold interval and geographical characteristics and draws the following research conclusions. Firstly, there is a significantly positive double-threshold effect between FDI and haze pollution; that is, whether FDI is at the first or second threshold, its influence on haze pollution is significantly positive, and its influence reaches the most substantial level within the two threshold values. Meanwhile, there is a significantly negative single-threshold effect between CSR and haze pollution, that is, the effect of CSR on haze pollution on both sides of the single threshold has the positive effect of increasing marginal efficiency. The management of haze pollution in China is accompanied by optimizing the FDI structure and improving CSR. However, those provinces where each variable crosses different threshold intervals have prominent geographical characteristics. Secondly, from the threshold value and interval distribution perspective, the number of provinces that are below the first threshold value of FDI and CSR decreases yearly. Improving the quality of FDI use and actively carrying out CSR activities have become new approaches to haze pollution control. In terms of the geographical distribution of provinces, the eastern region, with its superior geographical features and developed economy, acts as the main force that crosses the second threshold of FDI and the first threshold of CSR. Meanwhile, the central provinces in the critical period of industrial optimization and investment attraction primarily lie between the two thresholds of FDI.

The above empirical evidence suggests that high-quality FDI and CSR can be used as tools to achieve haze pollution control targets and to construct a green, low-carbon and circular economic system. Policy recommendations are then proposed as followed: Firstly, the quality and structural optimization of FDI should be given priority. Given the need for high-quality economic development, people should adhere to the environmental access threshold of FDI, reduce the entry of enterprises with high energy consumption and pollution, and introduce more clean production and technological innovation enterprises. A group of foreign enterprises that are equipped with technological advantages and are in line with China’s economic development should also be introduced to reduce the probability of haze pollution and its negative effects by accumulating and diffusing their capital, technology and knowledge. Secondly, the fulfilment of social responsibilities should be vigorously promoted. Considering their current situation in fulfilling their social responsibilities, foreign and domestic enterprises should be guided to form a development concept that combines high-quality economic development with ecological and environmental protection. These enterprises should also jointly design social responsibility projects with the government and the public to solve social and environmental problems. They should extend their social responsibility to the whole industrial chain to form standard social and environmental value norms, drive chain enterprises to participate in social and environmental governance practices, and coordinate and cooperate with one another to address haze pollution. Thirdly, an environmental governance system shared amongst the government, enterprises and society should be established. The government should not only share its responsibility through environmental regulations and strengthen the restraint mechanism of enterprises’ pollution emissions, but also promote a long-term cooperation mechanism with enterprises, social organizations and other actors in environmental governance by taking advantage of the situation. Enterprises should also take an active part in this process by appropriately increasing their R&D investment to innovate green and clean production technologies and by exploring and developing closed-loop value creation systems that reduce emissions and costs, save production materials and recycle energy in circulation. They can significantly contribute to improving the environment by working with third parties, such as universities and research centers, in developing joint business plans, such as eco-patent sharing, to expand the space for collaboration.

## Figures and Tables

**Figure 1 ijerph-20-04802-f001:**
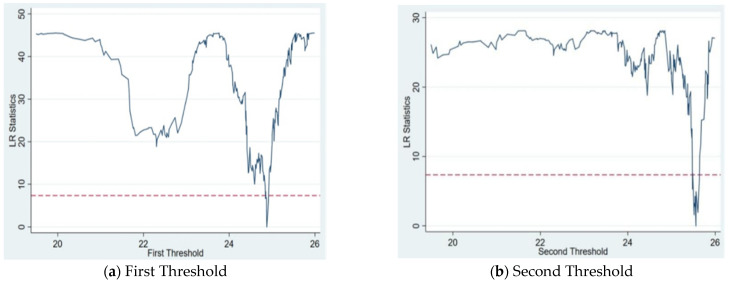
Plot of threshold estimates and likelihood ratio function for FDI.

**Figure 2 ijerph-20-04802-f002:**
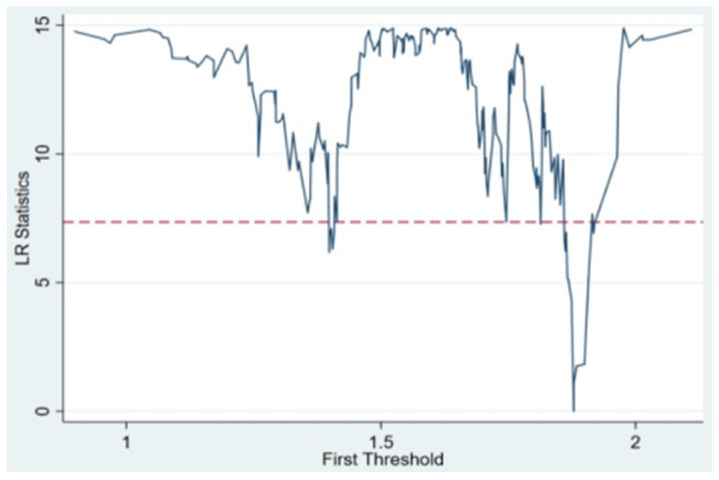
Plot of the threshold estimate of CSR versus the likelihood ratio function.

**Table 1 ijerph-20-04802-t001:** Correlation and variance inflation factors of variables.

Variable	PM	FDI	CSR	EG	ES	IS	ER	VIF	1/VIF
PM									
FDI	0.456 *							2.32	0.432
CSR	0.183 *	0.151 *						1.32	0.756
EG	0.403 *	0.470 *	0.469 *					2.19	0.457
ES	−0.213 *	−0.246 *	0.083	0.108				1.19	0.838
IS	−0.085	−0.207 *	−0.245 *	−0.476 *	−0.215 *			1.41	0.712
ER	−0.173 *	−0.484 *	0.042	−0.190 *	0.078	0.045		1.73	0.729
IA	0.401 *	0.676 *	0.186 *	0.592 *	−0.056	−0.388 *	−0.402 *	2.41	0.415

Note: * Indicates significance at the 10% level.

**Table 2 ijerph-20-04802-t002:** Regression results of FDI, CSR and Haze Pollution.

Variable	Haze Pollution							
	Coefficient	Standard Error	t	*p*	Coefficient	Standard Error	t	*p*
FDI	−25.560	13.908	−1.84	0.067	3.614	0.774	4.67	0
FDI^2^	0.645	0.305	2.11	0.035				
CSR	−13.569	4.745	−2.86	0.005	31.601	29.245	1.08	0.281
CSR^2^					−15.202	9.166	−1.66	0.098
EG	−0.628	3.005	−0.21	0.835	−0.412	3.010	−0.14	0.891
IS	5.441	11.165	0.49	0.626	5.194	11.212	0.46	0.644
ES	−38.968	12.402	−3.14	0.002	−42.179	12.490	−3.38	0.001
ER	4.910	3.311	1.48	0.139	5.093	3.318	1.53	0.126
IA	0.247	0.130	1.90	0.058	0.366	0.121	3.04	0.003
R^2^(F)	0.328 (17.20)				0.324 (16.89)			

Note: *p* < 0.1 indicates significance at the 10% level, *p* < 0.05 indicates significance at the 5% level, *p* < 0.01 indicates significance at the 1% level.

**Table 3 ijerph-20-04802-t003:** Test of FDI and CSR threshold effect results.

Independent Variable	Threshold Variable	Thresholds	F	*p*	Threshold Value	95% Confidence Interval	Critical Value		
							1%	5%	10%
FDI	FDI	Single	22.02	0.000	24.877	(24.844, 24.900)	28.868	26.807	25.941
		Double	24.74	0.000	25.558	(25.511, 25.561)	14.928	12.926	11.618
		Triple	23.88	1.000	22.302	(21.971, 22.318)	33.099	29.304	27.418
CSR	CSR	Single	17.67	0.080	1.879	(1.826, 1.879)	10.486	9.971	9.026
		Double	16.05	0.3200	1.976	(1.966, 1.988)	9.867	8.646	7.692

Note: *p* < 0.1 indicates significance at the 10% level, *p* < 0.05 indicates significance at the 5% level, *p* < 0.01 indicates significance at the 1% level.

**Table 4 ijerph-20-04802-t004:** Threshold effect model regression results.

Variable	Haze Pollution							
	Coefficient	Standard Error	t	*p*	Coefficient	Standard Error	t	*p*
FDI					3.536	0.768	4.61	0.000
CSR	−11.947	4.236	−2.82	0.005				
EG	−1.325	2.741	−0.48	0.629	−1.254	3.016	−0.42	0.678
IS	15.005	10.3501	1.45	0.148	4.863	11.122	0.44	0.662
ES	−36.896	11.370	−3.25	0.001	−43.796	12.414	−3.53	0.000
ER	4.107	3.054	1.34	0.180	5.153	3.290	1.57	0.118
IA	0.491	0.130	3.79	0.000	0.422	0.122	3.45	0.001
FDI-1	1.689	0.760	2.22	0.027				
FDI-2	2.273	0.730	3.11	0.002				
FDI-3	1.637	0.749	2.18	0.030				
CSR-1					−9.652	5.199	−1.86	0.064
CSR-2					−14.040	4.629	−3.03	0.003
F	24.74				17.67			
R^2^	0.442				0.334			

Note: FDI-1, FDI-2, and FDI-3 refer to low-, medium-, and high-intensity intervals of FDI; CSR-1 and CSR-2 refer to the low- and high-intensity intervals of corporate social responsibility, respectively; *p* value is the result obtained by repeated sampling 300 times with Bootstrap.

**Table 5 ijerph-20-04802-t005:** Statistical results of the number of provinces within different threshold intervals, 2009–2018 (unit: one).

Threshold Interval	2009	2010	2011	2012	2013	2014	2015	2016	2017	2018
FDI ≤ 24.877	15	24	22	22	21	21	20	18	18	18
24.877 < FDI ≤ 25.558	9	3	5	5	6	6	7	10	9	9
FDI > 25.558	6	3	3	3	3	3	3	2	3	3
CSR ≤ 1.879	30	30	30	29	29	26	25	25	23	26
CSR > 1.879	0	0	0	1	1	4	5	5	7	4

## Data Availability

The data presented in this study are available on request from the corresponding author. The data are not publicly available due to student confidentiality and privacy regulations.

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
