# Peer review of "Research on the Heterogeneity Threshold Effect of Foreign Direct Investment and Corporate Social Responsibility on Haze Pollution"

_ijerph, 2023, doi:10.3390/ijerph20064802_

Round 1
Reviewer 1 Report
This manuscript addressed an important issue and provide new approach of the impact of foreign direct investment and corporate social responsibility on haze pollution and their nonlinear relationship. Follow questions should be answered.
(1) Abstract: Please describe in the second or third sentence about the shortcomings or progress the previous research of impact of foreign direct investment and corporate social responsibility on haze pollution.
(2) Methodology and data: Please move the 'Theoretical mechanisms and research hypotheses' section from literature review to this section
(3) Discussion: Please add a separate Discussion Section to make comparison with the peer studies of the relations between FDI, CSR and haze pollution
(4) Conclusions: Besides the policy suggestions, please add a summary of the theoretical significance of this paper.
Author Response
(1) Abstract: Please describe in the second or third sentence about the shortcomings or progress the previous research of impact of foreign direct investment and corporate social responsibility on haze pollution.
A1: Thanks for your instructive suggestions. According to your comments, we have checked and rewritten the abstract, in which the shortcomings of the previous research on the impact of foreign direct investment and corporate social responsibility on haze pollution have been added in the revised manuscript. Please see the revised manuscript.
(2) Methodology and data: Please move the 'Theoretical mechanisms and research hypotheses' section from literature review to this section.
A2: Thanks for your valuable suggestions. According to your comments, we have moved the 'Theoretical mechanisms and research hypotheses' section from literature review to this section. Please see the first paragraph of Section‘2. Methodology and data’in the revised manuscript.
(3) Discussion: Please add a separate Discussion Section to make comparison with the peer studies of the relations between FDI, CSR and haze pollution.
A3: Thanks for your comments. We have added a separate Discussion Section to make comparison with the peer studies of the relations between FDI, CSR and haze pollution. Please see the Section ‘4. Discussions’ in the revised manuscript.
(4) Conclusions: Besides the policy suggestions, please add a summary of the theoretical significance of this paper.
A4: Thanks for your comments. According to your comments, we have added a summary of the theoretical significance of this paper. Please see the seventh paragraph of Section ‘1. Introduction’ in the revised manuscript.

Reviewer 2 Report
Dear authors,
The article lacks a clearly formulated scientific problem or problematic questions. Once the scientific problem has been formulated, the results obtained will need to be adjusted accordingly. Scientific problem should be added to abstract as well. Please look to list of references more detailed, now in the text there are 41 scientific source, but in list of references only 40.
Author Response
The article lacks a clearly formulated scientific problem or problematic questions. Once the scientific problem has been formulated, the results obtained will need to be adjusted accordingly. Scientific problem should be added to abstract as well. Please look to list of references more detailed, now in the text there are 41 scientific source, but in list of references only 40.
A: Thanks for your careful review of our manuscript. According to your comments, we have rewritten the Section ‘1. Introduction’. Specially, we have described in more detail the refined subjects involved in the issues studied in this paper, such as the shortcomings or progress of the previous research on the impact of foreign direct investment and corporate social responsibility on haze pollution, etc. Finally, we determined the scientific issue of this study: to explore the phased impact of FDI and CSR on haze pollution using threshold effect model. Meanwhile, we have checked and rewritten the abstract, added the Scientific problem in it. Please see the revised manuscript. Also, we have carefully added and reorganized the Section ‘Discussions’ based on the refined results in the revised manuscript, according to your comments. Finally, thanks so much for helping us point out some low-level errors in the format of the references. We have double checked the references by ourselves and also revised it, conscientiously. Please see the revised manuscript.

Reviewer 3 Report
Overall it is good analysis and approach. However, I have few comments and suggestions to further improve the draft. My comments and suggestions are following:
- Revise abstract for the conciseness, clarity, and uniqueness. Thus far it is not a standard abstract. Add practical implications at the end. Moreover, also highlight significant results.
- Useless to add literature review section in such a technical paper. Why need arises? Better to delete or merge this section with the introduction and write story type section as a single.
- How and why to rely on given data sources? How and who will ascertain that those data sources were really obtained and utilized in its true sense?
-Why this time period was selected?
-Why panel regression model was used? why not others?
- Same comments for conclusions as for abstract or so.
- I am not convinced by the way results presented. Discussion part should be there. section 5.5 is merely low.
-Better to add a separate discussion section.
-Acknowledgement section is incorrect.
-Look for any redundancy of the references.
Author Response
(1)- Revise abstract for the conciseness, clarity, and uniqueness. Thus far it is not a standard abstract. Add practical implications at the end. Moreover, also highlight significant results.
A1: Thanks for your instructive suggestions. According to your comments, we have checked and rewritten the abstract, in which practical implications and significant results are added. Please see the revised manuscript.
(2)- Useless to add literature review section in such a technical paper. Why need arises? Better to delete or merge this section with the introduction and write story type section as a single.
A2: Thanks for your valuable suggestions. Because the novelty of this paper should be further justified by highlighting main contributions by the existing literature, helping us to further clarify the motivations and contributions of this paper. So we merged this section with the ‘Introduction’, added a specific example that explicitly states FDI, economic aggregate and environmental pollution time-variation of Beijing-Tianjin-Hebei Region during the period of 2009-2018. please see the Section ‘Introduction’ and marked in red in the revised manuscript.
(3)- How and why to rely on given data sources? How and who will ascertain that those data sources were really obtained and utilized in its true sense?
A3: Thanks for your comments. Because given data resources are more accurate and scientific, some data cannot be obtained only by themselves. For example, the data released by the National Statistical Yearbook need to be completed with a lot of human and financial resources in the process of obtaining. Therefore, such data is more convincing and credible. For another example, at present, there are two main databases for corporate social responsibility evaluation data in China: RANKINS ESGRATINGS (RKS) and Dispatch net. Many scholars, such as Yu Hongyan (2015), Jing Longjiao (2020), Yang Xingzhe (2022), have published innovative articles in journals of Management Review, Statistics & Decision and China Journal of Economics using CSR data provided by RKS. In view of this, this study uses the given data source instead of the questionnaire data.
(4)-Why this time period was selected?
A4: Thanks for your comments. Since the COVID-19 broke out in China in 2019, some data could not be updated and released in time, so this paper chose the period of 2009-2018 for research.
(5)-Why panel regression model was used? why not others?
A5:Thanks for your comments. Panel data is two-dimensional data, one dimension is cross-sectional unit, and the other dimension is time series observations. The combination of cross-sectional data and time series data increases the sample size, and also considers the heterogeneity between different individuals, enterprises and regions. In addition, panel data reduces the collinearity between variables, which is more suitable for studying the dynamic changes of data, enhancing the technicality of mathematical analysis. In short, compared with the individual section data or time series data, the panel data make the analysis that could not be carried out possible.
(6)- Same comments for conclusions as for abstract or so.
A6:Thanks for your valuable suggestions. According to your comments, we have checked the relationship between abstract and conclusions, and further rewritten the abstract, in which the comments different from conclusions has been added in the revised manuscript. Please see the revised manuscript.
(7)- I am not convinced by the way results presented. Discussion part should be there. section 5.5 is merely low.
A7:Thanks for your instructive suggestions. We have carefully reorganized and rewritten the Section ‘Conclusions’ based on the refined results and discussions in the revised manuscript. ‘The Section 5.5’ is not further discussion and analysis but a further display of research results, according to your comments, we reflected this section as‘3.5. Further analysis’. Please see the revised manuscript.
(8)-Better to add a separate discussion section.
A8: Thanks for your comments. We have added a separate Discussion Section to make comparison with the peer studies of the relations between FDI, CSR and haze pollution. Please see the Section ‘4. Discussion’ in the revised manuscript.
(9)-Acknowledgement section is incorrect.
A9: Thanks for your careful review of our manuscript. Given that this paper does not involve confidentiality issues, the data used are open and transparent, and no administrative and technical support issues involved, according to your comments, we have get rid of this part.
(10)-Look for any redundancy of the references.
A10: Thanks so much for helping us point out some low-level errors in the format of the references. We have double checked the references by ourselves and also revised it, conscientiously. Please see the revised manuscript.

Round 2
Reviewer 3 Report
Changes are acceptable.